# OpenReview forum: "One Token to Fool LLM-as-a-Judge"
_ICLR.cc/2026/Conference — Submitted to ICLR 2026_

### Official Review · Reviewer_ivRn · 2025-10-17

**Soundness:** 3
**Presentation:** 3
**Contribution:** 2
**Rating:** 2
**Confidence:** 4

**Summary:**

This paper investigates the issue of reward hacking concerning LLM-as-a-Judge. They propose two kinds of MASTER KEYs, non-word symbols and reasoning openers. They found that these superficial inputs can fool LLM-as-a-Judge. This issue extends to a wide range of models. They propose a method of leveraging synthetic data to mitigate this issue. It demonstrates effectiveness. Agreement experiments are conducted.

**Strengths:**

- This paper studies an urgent research question.
- Rewarding experiments of human agreement.
- The idea of synthetic data is simple yet effective.
- The writing is easy to follow.

**Weaknesses:**

From my perspective, this paper is well-organized. My only concern is about the novelty.
- The MASTER KEYs are seemingly just another type of reward hacking patterns for LLM-as-a-Judge.
- This paper discovers some new patterns that will hack LLM-as-a-Judge, evaluate the issues on several models (not the most recent), SFT to correct these, and demonstrate the effectiveness of training. It is not exciting enough.

**Questions:**

- In line 294, the generalizability is emphasized. What about an overlap analysis between the training set and the test set?
- The observation that training on some kinds of known vulnerabilities can mitigate similar vulnerabilities is not beyond expectation. There are definitely other kinds of MASTER KEYs for LLM-as-a-Judge. As a starting experiment, can training on *Non-word symbols* help mitigate *Reasoning Openers*?
- As CoT is involved in LLM-as-a-Judge and reasoning models demonstrate better resilience, what about evaluating some new reasoning models, e.g., Qwen3-8B? Will adopting new open-source reasoning models alleviate this problem?
- What about the downstream implications concerning RL training? It will be more interesting to see whether correcting the limited set of MASTER KEYs brings about a better LLM.
- The paper title is somewhat misleading. It is unclear what the *One Token* refer to.

---

> ### Author Response · Authors · 2025-11-23
>
> Thanks for reading this work and providing feedback. We appreciate that this work addresses an urgent question, that the fix is simple yet effective, and that the presentation is easy to follow. Please find important clarifications to most important concerns shared by reviewers in the general response above. Below we answer to other questions.
>
>
> * Q:  In line 294, the generalizability is emphasized. What about an overlap analysis between the training set and the test set?
>
> A: As we mentioned in the general response, the training set and the test set have zero overlap. This generalization capability is a core contribution of our work.
>
>
> * Q:  The observation that training on some kinds of known vulnerabilities can mitigate similar vulnerabilities is not beyond expectation. There are definitely other kinds of MASTER KEYs for LLM-as-a-Judge. As a starting experiment, can training on Non-word symbols help mitigate Reasoning Openers?
>
> A: In our training set, it only contains the first sentence generated by GPT, so it is analogous to the reasoning opener attack but has zero overlap with the test case in Table 1. We can observe that training on these reasoning opener attacks can also mitigate the hack on non-word symbols.
>
>
> * Q:  As CoT is involved in LLM-as-a-Judge and reasoning models demonstrate better resilience, what about evaluating some new reasoning models, e.g., Qwen3-8B? Will adopting new open-source reasoning models alleviate this problem?
>
> A: We have tested Qwen3-8B, and the experimental results are provided in the general response above. Please check that part. We find that Qwen3-8B also exhibits high false-positive rates, and the thinking mode is even worse than the non-thinking mode, which aligns with our findings in Section 3.6 that more thinking may even exacerbate this problem rather than alleviate it.
>
>
> * Q:  What about the downstream implications concerning RL training?
>
> A: As shown in Figure 2, using standard LLM-as-a-judge can lead to policy collapse due to the hack. When reward model hacking happens, the RL-based post-training would also fail. However, when using our Master-RM on the same setup, we did not observe such a collapse, and the policy stably improves.
>
>
> * Q:  The paper title is somewhat misleading. It is unclear what the One Token refer to.
>
>
> A: The One Token refers to the Non-word symbols in Table 1, where we show that even a single token like “:” can induce high false positive rates in some of the most advanced large language models.

---

> ### Comment · Reviewer_ivRn · 2025-11-23
>
> Thanks for the authors' responses.
>
> LLM-as-a-Judge (or the reward model referenced in this work) can be compromised by tricky answers. This is not new knowledge for the community. I still have several persistent concerns and questions that remain unresolved.
> - This work mainly focuses on two types of MASTER KEYs: Non-word Symbols and Reasoning Openers. These concrete attack exposures are valuable, but they fall short of constituting substantial contributions for a top-tier conference. What we truly need is a method to continuously discover new attacks. For identifying new MASTER KEYs, we care more about new types of attacks rather than the new attack instances demonstrated in Section 3.5. Is there any relevant exploration into discovering new attack types beyond Non-word Symbols and Reasoning Openers?
> - The manuscript repeatedly emphasizes the importance of a robust reward model for RLVR, which is certainly correct. However, it raises more curiosity about the extent to which the varying FPRs measured in this work translate to improved model performance after RLVR. The current manuscript mentions only one RL run in Figure 1, which is far from sufficient to answer this question.

---

> ### Author Response · Authors · 2025-11-26
>
> We thank the reviewer for the follow-up. Below we make further clarifications
>
> 1. Scope of this work
>
> We emphasize that "Reasoning Openers" and "Non-word Symbols" are not isolated instances, but representative symptoms of a systemic vulnerability to **"Semantic Vacuity."** To address the need for automatic hacking discovery, Section 3.5 presents a generalized, automated framework for mining the embedding space. This demonstrates that our approach effectively identifies unseen adversarial clusters beyond the specific 10 examples tested in Table 1.
>
> We acknowledge that new attack vectors may emerge in real-world settings. However, robustness research advances through an iterative **"discover-and-mitigate" cycle**. While no defense can guarantee immunity against all unknown future attacks, our contribution is to identify and mitigate a **concrete, widespread vulnerability** affecting nearly all SOTA judges.
>
> Moreover, both the anti-hacking results (Table 1) and the judge models' evaluations (Table 3) demonstrate that our Master-RM models are not only robust but also performant. Our Master-RM-32B not only shows lower false-positive rates but also achieves impressive accuracies/macro F1 scores of 95.15%/95.14% and 86.80%/81.96% on the two public, verifiable benchmarks, respectively. These scores surpass all open-source models and are highly competitive with leading closed-source models, outperforming other general-domain, widely used LLM judges such as GPT-4o, GPT-4o-mini, and Claude-4-Sonnet. Obtaining such performant, open-sourced, general-domain generative reward models is also one of our major contributions.
>
> 2. On RLVR Performance (Clarification on Stability vs. Optimization)
>
> The reviewer asks how reduced FPR translates to improved model performance. We believe the reviewer does not fully understand the scope of this work. We wish to clarify that our method does not aim to improve RLVR performance by a certain percentage (optimization), but rather to prevent catastrophic failure (stability).
>
> As shown in Figure 2, without resistance to these attacks, the policy collapses to producing trivial, length-zero responses, effectively killing the training.
>
>  Consequently, the translation of FPR to performance is not linear (e.g., "+5% math reasoning accuracy"), but binary: the difference between a training run that collapses and one that functions correctly. Our Master-RM is a necessary safeguard to enable RLVR to work at all.

---

### Official Review · Reviewer_U6so · 2025-10-18

**Soundness:** 2
**Presentation:** 2
**Contribution:** 2
**Rating:** 4
**Confidence:** 3

**Summary:**

The paper reveals a significant vulnerability in the "LLM-as-a-Judge" paradigm via systematic evaluations. Specifically, the author find most reward models would give high rewards to superficial inputs with the master keys which include non-word symbols or generic reasoning openers. To address this threat, the authors propose a simple yet effective data augmentation method to fine-tune the reward models. The resulting models (i.e. master RMs) are not effective in defending master-key attacks but also exhibit good performance on verification benchmarks.

**Strengths:**

1. This paper discusses a timely and critical topic for LLM-as-a-judge.
2. The paper proposes a simple hacking method and conducts a broad evaluation on the robustness of different reward models against "master key" attacks
3. The paper proposes a simple and yet effective data augmentation method to defend the master key attack.

**Weaknesses:**

1.  Although the paper provides many interesting findings, it lacks an in-depth analysis of the model vulnerabilities. For example,

a. Why "master key" attacks can succeed in fooling General-Purpose LLMs, i.e. what training behavior could lead General-Purpose LLMs to give high reward to "master key" attacks?

b. According to Table 1, multilingual reasoning openers also achieve a high false positive rate, despite the fact that the evaluation tasks are English-based. Why do reward models give high rewards to the generation in a different language?

2. More experiments should be done to verify the effectiveness of master-reward models.

a. The author should consider the adaptive attacks where the attackers aim to further hack the master-reward models, e.g. the other attack methods discussed in "Vulnerabilities of LLM-as-a-judge"  or the new master key patterns found in Appendix C.

b. It is good to show if applying master-reward models during RL training can improve the training performance compared to other reward models.

3. The paper is not well organized and needs improvement:

a. Section 3 " Methodology" discusses Verifiable Reward Modeling in RLVR, which neither belongs to the methodology nor is the main focus of the paper.

b. The methodology part is mixed with the experiment setup in Section 4. The author should move the description of "Master-RM" to Section 3.

c. Section 4.3.2 and Section 4.3.3 seem to serve a duplicate purpose and they should be merged together. Besides, Section 4.3.2 lacks the details for how to get human judgments.

d. Additional Experimental Results part(Line 467) only lists the findings without a detailed discussion. It is important to make the main paper self-contained and move some of the discussion to the main paper.

**Questions:**

See above. Besides, I have some additional questions:

1. In Line 297, the authors claim that master RMs can generalize to unseen hacking attacks as well. Indeed, the data augmentation strategy seems not to include multilingual opening reasoners, but still achieves zero false positive rate for them. Could you explain the insights behind this phenomenon?

2. According to Table 3, master RMs also show superior performance over specialized RMs on general verification benchmarks (i.e. VerifyBench). Is it due to using a more powerful training dataset? The authors may need to compare with the performance of training the same base model with the same dataset without data augmentation.

---

> ### Author Response · Authors · 2025-11-23
>
> We thank the reviewer for reading this work and for appreciating that it studies a critical topic in LLM-as-a-Judge and has proposed a simple yet effective strategy to address an important weakness. We have provided clarifications on some of your concerns in the general response above. Below, we respond to other questions.
>
> * Q: In Line 297, the authors claim that master RMs can generalize to unseen hacking attacks as well. Indeed, the data augmentation strategy does not seem to include multilingual reasoning openers, yet the model still achieves a zero false positive rate for them. Could you explain the insights behind this phenomenon?
>
> A: As we mentioned in our common response, the training and test sets have zero overlap, so all test cases in Table 1 are unseen during training. The augmented data in the training set provide negative examples to teach the model that plausible reasoning openers do not contain useful information and therefore should be treated as incorrect response. This encourages the model to ignore superficial, potentially meaningless tokens and to focus instead on whether the core reasoning and final answer are correct. Such behaviour can generalize beyond the training set.
>
> * Q: According to Table 3, master RMs also show superior performance over specialized RMs on general verification benchmarks (i.e., VerifyBench). Is this due to using a more powerful training dataset? The authors may need to compare with the performance of training the same base model on the same dataset without data augmentation.
>
> A: In Table 3, the “Multi-sub RM” row corresponds to a model trained with the same base model and the same dataset as Master-RM-7B, but without our data augmentation. Thus, the comparison between Multi-sub RM and Master-RM-7B isolates the effect of the augmentation. As shown in the table, Master-RM-7B performs slightly better on average. We hypothesize that reducing hallucinations on reasoning openers also lowers the standard false positive rate: when a reasoning path contains an incorrect final answer but includes persuasive reasoning openers, a vanilla judge may be hacked by these openers. Our augmentation strategy explicitly trains the model to discount such patterns, which in turn improves robustness on general verification benchmarks.

---

### Official Review · Reviewer_ua7e · 2025-10-28

**Soundness:** 2
**Presentation:** 3
**Contribution:** 2
**Rating:** 4
**Confidence:** 4

**Summary:**

LLM-judges are often tricked by generic tokens (e.g. a colon or full-stop) that leads to false positive results, affecting models such as o1 and Claude. To alleviate this problem, the paper propose a data augmentation strategy with truncated model outputs as negative examples. The resulting models trained with the augmented data maintains high performance in standard evaluations while showing SOTA robustness against these generic token attacks.

**Strengths:**

1.	Given that LLM-Judges are increasingly used for training and evaluation, highlighting a flaw that LLM Judge have can be important. It is interesting how many of these master keys are so generic e.g. a blank space or a colon can substantially change results. The is also related to the topic of jailbreaking LLMs to do things that they were not supposed to, and can serve as a bridge between this LLM-judge and jailbreaking.

**Weaknesses:**

1.	The paper mostly discusses False Positive Rate, which is a key issue of model-based verifiers. However, it’s also important to discuss False Negative Rate since we should not have models that have zero false positive rate, by simply predicting all generated answers as wrong. It’s important to measure and report both since rule-based verifier are known to be low in false-positives and high in false-negatives compared to model-based verifiers [1]. Both can be combined into an overall F1 metric.
2.	The augmentation approach seems a little under-explored as it only takes the first sentence regardless of its content (there doesn’t seem to be any method to check if the first sentence does indeed have useful content, besides an assertion in line 272). This encourages the judge model to ignore the generated solution and increases the likelihood of “NO” compared to “YES”, which can reduce false positives, but will likely also increase false negatives. The other risk of this approach is that it will over-penalize any response that has a similar opener as the data-augmentation model (GPT-4o-mini) rather than solve this behavior.
3.	The paper does very limited exploration on why the current LLM-Judges are easily hackable by the master-keys. Do those master keys encourage the model to first generate its own answer rather than to just judge on the provided answers?
4.	The empirical performance of Master-RM in Table 2 doesn’t seem to be better than the original Multi-sub RM, which the Master-RM training data is primarily derived from. This suggests that the data augmentation doesn’t improve the RM quality in terms of its agreement with either humans or GPT-4o (a standard LLM-Judge).

[1] From Accuracy to Robustness: A Study of Rule- and Model-based Verifiers in Mathematical Reasoning

**Questions:**

1.	What’s Claude-4 - Is that Claude Sonnet 4 or Claude Opus 4?
2.	Can the authors elaborate upon how the agreement with human annotations data was collected? Specifically, is the task challenging for these annotators and how much do the annotators agree with each other?
3.	I see the prompt template in Table 6 – what happens when the generated answer is neither YES nor NO? Also, how many tokens is the LLM-judge allowed to generate? I assume it’s only 1 token but it is explicitly stated here.

---

> ### Author Response · Authors · 2025-11-23
>
> We thank the reviewer for reading this work and providing important comments. Some of your concerns have been addressed in the general response above, please review. We answer other questions below.
>
> * Q: What’s Claude-4 - Is that Claude Sonnet 4 or Claude Opus 4?
>
> A: As shown in Appendix B.1 (Table 7), "Claude-4" refers to Claude 4.0 Sonnet (specifically version 20250514).
>
> * Q: Can the authors elaborate upon how the agreement with human annotations data was collected? Specifically, is the task challenging for these annotators and how much do the annotators agree with each other?
>
> A: The annotation difficulty varies significantly across datasets: while MATH and GSM8K are often straightforward (checking values or expressions), Multi-subject RLVR and Natural Reasoning involve complex, domain-specific long-form explanations (e.g., Physics, Astronomy) that require careful expert verification. Therefore, rather than aggregating independent annotator votes, we employed a consensus-based approach among the authors. We manually and jointly analyzed each response, discussing and resolving ambiguous cases to establish a definitive "human gold standard" for every question.
>
>
> * Q: I see the prompt template in Table 6 – what happens when the generated answer is neither YES nor NO? Also, how many tokens is the LLM-judge allowed to generate? I assume it’s only 1 token but it is explicitly stated here.
>
>
> A: Since some models generate intermediate reasoning before producing a final YES or NO, we set a uniform token budget of 1024 for all open-source models. For each generation, we extract the last line of the model’s output and parse it to obtain the predicted YES or NO. In our implementation, generations from which a valid YES or NO cannot be parsed from the final line are discarded. The ``Success of Parsing’’ rate in Table 2 reports the fraction of generations for which a valid YES or NO is successfully extracted. This rate is nearly 100% across all evaluated models.

---

> > ### Comment · Reviewer_ua7e · 2025-11-24
> > **Reply to author response**
> >
> > Question 1: please don't leave critical information like the actual model name to the appendix in the future. Instead, please use the actual model names in the main paper and tables.
> >
> > Question 2: Can you clarify how `manually and jointly analyzed each response` was carried out? It sounds here like the annotators sat in a room to go through the answers side by side but in my experience of doing data collection, the most typical pipeline is to have each annotator do it separately and then when there's disagreement, discuss. If so, what was the initial rate of disagreements? How did the authors ensure that the annotators had sufficient expertise in all of the topics in Multi-RVLR / AIME , which as the authors say require experts to verify. It's also slightly confusing here but are the annotators formed by the authors? This sentence ` we employed a consensus-based approach among the authors` suggest that this is the case.
> >
> > Weakness 1: ` Our Master-RM-32B achieves an average of 95.15%, outperforming most SOTA models. This score would be mathematically impossible if our models were "over-penalizing" correct answers (i.e., if it had a high FNR).` : I disagree that this is mathematically impossible. It is very much possible if the data is highly imbalanced.
> >
> > Weaknesses 2-4: I don't see these weaknesses addressed either in the general or reviewer-specific response.

---

> ### Author Response · Authors · 2025-11-25
>
> We thank the reviewer for the quick response. We have updated our paper again: we added the information of Claude-4 in the main text and updated Table 3 with detailed F1 scores. Your questions are elaborated below.
>
> **Q1:**  Clarify how manually and jointly analyzed each response.
>
> **A:**  You are right, we let two persons to score the same question separately and if they disagree with each other, we introduce the third person to discuss. The initial agreement rate achieved a Cohen’s kappa of 0.9596.  We are happy to provide detailed evaluation data if the reviewer would like to know. For math dataset like AIME, we can simply compare the answer in a response with the reference answer. For general reasoning dataset like Multi-RLVR, it contains questions in different fields of science that has reference answer but is sometimes long, we evalute whether the response of model shares the same meaning as the reference answer. This requires some background knowledge to evaluate. Our annotators are all PhD students in computer science and statistics with general education in undergradute-level physics, social science and other fields. With search engines, they have the ability to check whether two answers have the same meaning for questions in this dataset.
>
>
> **Q2:**  Concerns on over-penalization,  false-negative rates and overall F1 metric.
>
> **A:**  We have updated our Table 3 in the latest PDF, showing that our reward model also has high F1 scores. Regarding the dataset structure, in VerifyBench the numbers of correct/incorrect instances are 1000/1000, and in VerifyBench-Hard they are 291/709.
>
> For completeness, we report the detailed true positive (TP), true negative (TN), false positive (FP), and false negative (FN) counts, together with Macro F1, for GPT-4o, master-rm-7B, and multi-sub RM below:
>
> | Model        | Dataset          | TP  | TN  | FP | FN  | F1    |
> |-------------|------------------|-----|-----|----|-----|-------|
> | GPT-4o      | VerifyBench      | 919 | 964 | 36 | 81  | 94.15 |
> | GPT-4o      | VerifyBench-Hard | 153 | 690 | 19 | 138 | 77.94 |
> | master-rm-7B| VerifyBench      | 924 | 965 | 35 | 76  | 94.45 |
> | master-rm-7B| VerifyBench-Hard | 210 | 634 | 75 | 81  | 80.98 |
> | multi-sub RM| VerifyBench      | 937 | 963 | 37 | 63  | 95.00 |
> | multi-sub RM| VerifyBench-Hard | 195 | 630 | 79 | 96  | 78.42 |
>
> Master-rm-7B uses the same training data as multi-sub RM, with an additional data-augmentation step. We observe that master-rm-7B produces fewer false positives (FP) than multi-sub RM on both VerifyBench and VerifyBench-Hard. For false negatives (FN), master-rm-7B is higher on VerifyBench (+13 compared with multi-sub RM), but lower on VerifyBench-Hard (−15 compared with multi-sub RM). Overall, when aggregating across both datasets, master-rm-7B also has a lower total number of false negatives while maintaining high F1.
>
> **Q3:** Do those master keys encourage the model to first generate its own answer rather than to just judge on the provided answers?
>
> **A:** In our experiments, we found that Claude-4 Sonnet often ignores the judging-only instruction: it first solves the question by itself and then compares the answer it derives with the reference answer, ignoring the provided candidate answers. This behavior significantly increases the false positive rate. However, this behavior is rare for other models, which typically only output “YES” or “NO” without additional text. A more thorough investigation of this mechanism is an interesting direction for future work.
>
>
> **Q4:** The empirical performance of Master-RM in Table 2 doesn’t seem to be better than the original Multi-sub RM, which the Master-RM training data is primarily derived from. This suggests that the data augmentation doesn’t improve the RM quality in terms of its agreement with either humans or GPT-4o (a standard LLM-Judge).
>
> **A:** Our data-augmentation method is specifically designed to mitigate reward hacking, rather than to improve performance on standard judging tasks. On these standard tasks, our model remains comparable to the best existing model, while being more robust. In Table 3, our Master-RM-7B even has slightly better perforamnce than Multi-sub RM. Thus, the augmentation improves robustness to master keys without sacrificing (and in some cases slightly improving) the general judging ability.

---

> > ### Comment · Reviewer_ua7e · 2025-11-28
> > **Followup reply to author response**
> >
> > Thank you for conducting the follow up analysis and details.
> >
> > After considering the response, I think the paper will benefit from one more round of major edits and peer review, beyond the discussion period. Therefore, I will maintain my current score.
> >
> > Some general comments:
> >
> > 1. Please try not to mix different metrics across the paper (the paper starts with FPR in Figure 1 and Table 1; then cohen's kappa in Table 2 - typically when matching model results with human results, the inter-rater agreement metric is not super useful; then accuracy/macro-F1 in Table 3). This is extremely confusing for readers. I recommend sticking to one metric (recommend macro-F1 since it's more protective against imbalanced datasets).
> >
> > 2. The paper needs to have substantially more details on how human annotations was done. It needs to make clear whether the authors were the annotators and if not, how the annotators were recruited and compensated for their time. It also needs to clarify how annotators do their task (e.g. what does with search engine mean and were they allowed to use LLMs for their annotations). In many cases, having human data collection also requires a dedicated ethics statement.
> >
> > 3. Limited explanation of how hacking with master keys works (beyond Claude 4 Sonnet). Similarly, I see in Appendix C that there are multiple "speculations" on how different model size behavior, which can be very interesting but further evidence is needed to develop this point into one contribution.
> >
> > 4. More focus in paper. I think the paper aims to do too many things, which makes it potentially confusing. For instance, the title `One Token to Fool LLM-as-a-Judge` suggests that it's a paper that only focussed on exploring what this one token is (and presumably the mechanism behind it) but almost half the paper is dedicated to the solution to overcome this. This isn't a weakness in itself but the way that it's presented currently makes it hard to follow.  Page 7 is also just filled by one big table now - for better clarity, might be better to have just two representative datasets in main paper and others in appendix.

---

### Official Review · Reviewer_HC5N · 2025-11-01

**Soundness:** 3
**Presentation:** 3
**Contribution:** 2
**Rating:** 2
**Confidence:** 5

**Summary:**

In this paper, authors investigate the fragility of LLMs being used as judges in RLVR for judging model response or in general LLM as a judge setup to evaluate model response. Something as simple as trivial tokens or superficial phrases (e.g., “Thought process:”
or “Let’s solve this problem step by step.”) are enough to break it ie the judge gives a false positive reward when the answer is meaningless. Effects hold through multiple models both general purpose LLMs or specialized RM. The proposed adversarial data augmentation strategy to mitigate this behavior.

**Strengths:**

* The authors highlight an important cautionary tale which the community should be aware of, especially with the increasing use of LLM as  judges in both RLVR and for evaluation purposes.
* The fact that this even happens in SOTA models is interesting. This will raise awareness in the community to take the LLM as a judge. Evaluations with a grain of salt.
* Clear presentation with an easy quick fix proposed

**Weaknesses:**

* The adversarial fix proposed is not thorough enough and would only guard against a very specific type of hacking of the judges.
* The solution approach is only tested against other RMs/LLMs on False positive rates where it does better. Its not surprising since they train for exactly those examples so their RM is naturally robust. More important is how this Master-RM does when used as an LLM as a judge in a real RLVR setting. Its possible in real RLVR setting even this master-RM gets hacked by some other kind of "attack" or "naive" prompts which the authors didn't train against.
* Authors didn't talk about automated ways of discovering these prompts, nor do they detail how they found these "attacks". They can talk about literature focussed on that too.

**Questions:**

* Did the authors try any reasoning Gen-RMs which reason before giving a response. Maybe because they spend more tokens at inference time and are trained in that manner maybe they are better. Eg https://huggingface.co/nvidia/Llama-3_3-Nemotron-Super-49B-GenRM . I know the authors tested o1 but maybe training for judging is important
* Maybe the authors can investigate an automated way of mining such adversarial examples using RL similar to setup in the paper [1] . These examples will naturally arise in the context of RLVR training but if you can do adversarial red-teaming or RL then you can find a much broader variety as compared to a small subste discussed in the paper
*  In line 270 "regenerate model responses using chain-of-thought prompting with GPT-4o-mini". Why not generate these from a reasoning model as they are the ones used for RLVR and their behaviour might differ from CoT prompted 4o-mini



[1] https://arxiv.org/abs/2504.06141v2

---

> ### Author Response · Authors · 2025-11-23
>
> We thank the reviewer for the insightful review. Most of your concerns/questions are addressed in the general response, and we elaborate on the others below.
>
> * Q: In line 270 "regenerate model responses using chain-of-thought prompting with GPT-4o-mini". Why not generate these from a reasoning model as they are the ones used for RLVR and their behaviour might differ from CoT prompted 4o-mini.
>
> A: We did not test the data augmentation with the first sentence generated from reasoning models, but the first sentences of non-reasoning models are also valid to use because they usually begin the reasoning with little information. For example, we found that the first sentences of reasoning models are similar in format across responses, they share the form "Okay, let’s try to figure out the problem…'';  while non-reasoning models have a bit more diverse first sentence like "To solve this problem, we need…'" or "We first consider…'' or just rephrase the question.
>
> Moreover, we would like to clarify that RLVR is not only for reasoning models; many papers also use RLVR to enhance the performance of non-reasoning models (e.g., Qwen3-4B-Base).

---

> ### Comment · Reviewer_HC5N · 2025-11-26
>
> Thank you for the rebuttal. Quite a few of my concerns  remain. I would like to maintain my score.

---

### Author Response · Authors · 2025-11-23
**General Response: Clarifications on Generalization, False Negative Rates, and Automated Discovery**

We thank all reviewers for their time and for agreeing on the importance and timeliness of the "master key" vulnerability in LLM-as-Judges.

However, we note that several reviews were based on impressions that differ from our actual experimental setup. We would like to take this opportunity to clarify three critical aspects shared across the reviews.

1. **Clarification on Training Data Independence.** A primary concern raised by Reviewer HC5N and ivRn is the impression that our model performs well because "they train for exactly those examples.", and "only guard against a very specific type of hacking".
**We wish to clarify that there is zero overlap.** As detailed in Section 3.2 (Line 229-243), our data augmentation uses truncated sentences from GPT-4o-mini's chain-of-thought responses (e.g., "To solve the problem, we need to..."). The "master keys" used for testing (e.g., "Thought process:", "Solution") are completely distinct from this training data. Therefore, the near-zero FPR reported in Table 1 demonstrates generalization to unseen attacks, rather than memorization. This generalization capability is a core contribution of our work.

2. **Clarification on False Negative Rates (FNR) and Evaluation Metrics.** Reviewers ua7e and U6so expressed concern that our approach might "over-penalize" correct answers (i.e., high FNR). We respectfully point out that our evaluation metrics already account for FNR. In the latest revised draft, we have also added clarifications in Lines 289-293 to fully address this concern.

  - Section 3.3.2 (Table 3) evaluates models on VerifyBench. The reported metric is overall accuracy, which, by definition, penalizes both FPs and FNs. Our Master-RM-32B achieves an average of 95.15%, outperforming most SOTA models. This score would be mathematically impossible if our models were "over-penalizing" correct answers (i.e., if it had a high FNR).
   - Section 3.3.2 (Table 2) measures LLM judges' agreement with human judgments and GPT-4o. Our models achieve Cohen's kappa values of 0.90 and 0.91. A high kappa score requires agreement on both positive ("YES") and negative ("NO") labels. A model that simply defaults to "NO" (high FNR) would have a kappa score near zero.

These results (Tables 2 and 3) are direct, empirical proof that our Master-RMs maintain powerful general verification capability and do not sacrifice FNR to achieve significantly lower False Negative Rate (FPR).

3. **Clarification on Automated Attack Discovery.** Regarding the comment by Reviewer HC5N that we did not discuss automated discovery methods, we would like to direct attention to Appendix D, which is dedicated entirely to this topic. We describe an automated method using embedding similarity to mine new "master keys", with hacking rates reported in Table 18. To make this part more noticeable, in the newly updated draft, we also describe this method and its performance in the main text. Please refer to Section 3.5.

---

> ### Author Response · Authors · 2025-11-23
> **Addressing common questions**
>
> We also would like to answer common questions below.
>
> * Q: Does reasoning-enhanced generative models or generative reward models have better performance on this attack? Do newer models like the Qwen3 perform better?
>
>  A: The answer is no, and reasoning can sometimes even make the false positive rate (FPR) higher. In the newly updated draft, Section 3.6 empirically demonstrates that inference-time techniques (e.g., introducing chain-of-thoughts prompting) may even degrade performance. One LLM judge we have tested, General-Verifier, has also employed chain-of-thoughts prompting but also exhibit vulnerability to hacking.
>
> Although Reviewer HC5N suggested evaluating Llama-3.3-Nemotron-Super-49B-GenRM, we found that this model is designed to assess the quality of a single response or rank two responses. However, our task is fundamentally different: we require the model to decide whether a candidate answer is equivalent to a reference answer and output a binary judgment. We also test open-sourced trained verifiers: General-Verifier and Omni-Judge, designed for our reference-based task, and the results in Table 1 show that they also have a high FPR.
>
>
> We test Qwen3 8B as the reviewers suggest, and the results below show that Qwen3 8B also has high FPRs, and Qwen3 8B (thinking) is usually worse than Qwen3 8B (non-thinking), which validates our observations that thinking more can increase FPRs.
>
> | Dataset\Models          | Qwen3-8B (thinking) (Avg/Worst FPR) | Qwen3-8B (non-thinking) (Avg/Worst FPR) |
> |------------------|--------------------:|------------------------:|
> | AIME1983-2024    | 10.7 / 13.8         | **10.8 / 24.2**             |
> | AIME2025         | **6.7 / 10.0**          | 4.0 / 6.7               |
> | GSM8K            | **74.2 / 90.1**         | 36.9 / 67.2             |
> | MATH             | **55.9 / 70.4**         | 36.6 / 63.8             |
> | Multi-sub        | 25.1 / **69.3**        | **32.6**/ 65.8            |
> | NaturalReasoning | **30.9 / 60.2**        | 29.5 / 48.3             |
>
>
>
> * Q: About the scope of the "master keys" identified in this work.
>
> A: Reviewer HC5N expressed concern that our solution might not be "thorough enough" because it might be hacked by "some other kind of 'attack'" in a real RLVR setting. We respectfully note that robustness research typically advances through an iterative "discover-and-mitigate" cycle. Our work identifies a specific, previously overlooked vulnerability that triggers catastrophic training collapse (Figure 2) and proposes a targeted, generalizable, and data-efficient solution. While no defense can guarantee immunity against all hypothetical or unknown future attacks, our contribution is the discovery and mitigation of a concrete, widespread vulnerability that affects nearly all current SOTA judges, which is a necessary and significant step forward.
> We hope this clarifies the scope and impact of this work.

---

> ### Author Response · Authors · 2025-11-23
> **Recap of Contributions and Empirical Insights**
>
> We would like to emphasize the contributions of this paper below.
>
> We consider the **reference-based reward model** used in RLVR, where the task for the reward model is to assess whether a given answer is equivalent to a reference answer and output binary feedback. This is a new paradigm that is different from the standard reward model, which either outputs a continuous score or preference feedback of two responses.
>
> In this setup, we identify a fundamental vulnerability in current reference-based reward models used in RLVR, which can **severely disrupt standard RL training and lead to training collapse**. We conduct a systematic analysis of this failure mode across a wide range of models and datasets, yielding the following four non-trivial observations:
>
> * First, the vulnerability is pervasive. It appears **consistently across models of different families and sizes**, and even strong proprietary models exhibit pronounced failures (Table 1).
>
> * Second, **simply scaling up the judge model does not mitigate the issue**; in fact, the vulnerability often worsens. The scaling trend is consistent yet non-monotonic (Section 3.4), with mid-sized models providing the best balance between robustness to the attack and accuracy on standard judging tasks.
>
> * Third, **inference-time techniques**, such as chain-of-thought (CoT) prompting or majority voting, **do not reliably defend against the attack** and, in some cases, exacerbate the vulnerability (Section 3.6).
>
> * Fourth, **removing the question from the judge prompt** substantially **reduces susceptibility** to the attack (Section 3.7). However, we also note that this trick may not be appropriate for general reasoning tasks.
>
> Together, these findings provide **a clearer understanding of how to deploy generative reward models more reliably in RLVR**. Our results suggest that using a mid-sized model, avoiding chain-of-thought prompting, and omitting the question from the judge prompt when possible generally leads to more robust performance.
>
> * Finally, to further address the vulnerability, we introduce a data-augmentation strategy and **train a new reward model** that **significantly reduces susceptibility to the attack while maintaining state-of-the-art accuracy on standard judgment benchmarks**.
>
> We have updated this paper to better emphasize our contributions and various empirical insights.

---

### Meta-Review · Area_Chair_w8fe · 2025-12-27

**Summary:**

The paper gives a systematic evaluation of the LLMs interms of generative reward models. Four reviewers engaged into the review process.  All reviewers show negative scores towards the paper.  The following concerns are of important.
1) For reviewer HC5N,  the proposed method suffers from the insufficient adversarial fix, limited experimental settings and lack of method of finding attacks. For questions, the reviewer mainly focuses on why not generate these from a reasoning model as they are the ones used for RLVR and their behaviour might differ from CoT prompted 4o-mini whichshould be carefully addressed
2) For reviewer ua7e, the reviewer gives a negative score. The comments include inadequate metrics(W1), simple augmentation method(W2) and limited improvements(W4) , which should be addressed. For questions,  the details of human annonations(Q1) can be described to improve clarity.
3) For reviewer U6so, there are three key weaknesses such as lack of in-depth analysis,  lack of sufficient experiments. For questions, in Q2, the experimental results based on the same base model with the same dataset without data augmentation should be listed.
4) For reviewer ivRn, W2 is important to be addressed, i.e., this paper discovers some new patterns that will hack LLM-as-a-Judge, evaluate the issues on several models (not the most recent), SFT to correct these, and demonstrate the effectiveness of training. It is not exciting enough.

In summary, the authors have tried to address the concerns raised the reviewers. However,  the paper still suffers from the limitations such as insufficient adversarial fix, limited experimental settings, lack of in-depth analysis,  lack of sufficient experiments. Therefore, the paper is rejected.

**Reviewer Concerns:**

For reviewer  HC5N,  the reviewer has replied to the rebuttals that quite a few of concerns remain. In details, the author does not reply point to point which it is not easy to judge which concerns are addressed perfectly.
For reviewer ua7e, the reviewer has replied to the rebuttals.  Some general concerns are still needed to be improved. For example, the details of human annonations, introduction of more performance metirc such as macro-F1 should be considered seriously.
For reviewer U6so,  two questions may be addressed by the authors but more details may be listed.  the weaknesses still remain outstanding.
For reviewer ivRn, the questions are partially addressed. But the weaknesses of the novelty still remain outstanding.

**Reviewer Scores:**

In the rebuttal phase, 3 of the reviewers i.e., HC5N, ua7e and ivRn  has commented that they would not change their score.  After reading the rebuttals to U6so, it is unlike to change his score.

---

### Decision · Program_Chairs · 2026-01-26

Reject